# Study on the Construction of a Time-Space Four-Dimensional Combined Imaging Model and Moving Target Location Prediction Model

**DOI:** 10.3390/s22176375

**Published:** 2022-08-24

**Authors:** Junchao Zhu, Qi Zeng, Fangfang Han, Huifeng Cao, Yongxin Bian, Chenhong Wei

**Affiliations:** 1School of Electrical Engineering and Automation, Tianjin University of Technology, Tianjin 300384, China; 2Tianjin Key Laboratory for Control Theory and Applications in Complicated Systems, Tianjin University of Technology, Tianjin 300384, China

**Keywords:** machine vision, artificial neural network, spatio-temporal model, motion localization prediction

## Abstract

Time-space four-dimensional motion target localization is a fundamental and challenging task in the field of intelligent driving, and an important part of achieving the upgrade in existing target localization technologies. In order to solve the problem of the lack of localization of moving targets in a spatio-temporal four-dimensional environment in the existing spatio-temporal data model, this paper proposes an optical imaging model in the four-dimensional time-space system and a mathematical model of the object-image point mapping relationship in the four-dimensional time-space system based on the central perspective projection model, combined with the one-dimensional “time” and three-dimensional “space”. After adding the temporal dimension, the imaging system parameters are extended. In order to solve the nonlinear mapping problem of complex systems, this paper proposes to construct a time-space four-dimensional object-image mapping relationship model based on a BP artificial neural network and demonstrates the feasibility of the joint time-space four-dimensional imaging model theory. In addition, indoor time-space four-dimensional localization prediction experiments verify the performance of the model in this paper. The maximum relative error rates of the predicted motion depth values, time values, and velocity values of this localization method compared with the real values do not exceed 0.23%, 2.03%, and 1.51%, respectively

## 1. Introduction

The study of the spatial trajectory localization prediction of motion targets currently has great research significance and application value in the fields of the military, aerospace, and intelligent driving [1]. Taking intelligent driving as an example, the number of traffic accidents due to drivers’ driving behaviors has increased rapidly in recent years. Strengthening the driver’s ability to analyze the road conditions and assisting the driver to judge the traffic environment are important ways to solve this problem. To track a moving target accurately in real time, it is necessary to predict the trajectory and state of the moving target. Motion target location prediction refers to capturing and finding the motion target and predicting its position information at the next moment based on the analysis of its previous motion trend [2]. In the field of intelligent driving, research on this problem is of great significance. In the process of car travel, for a sudden obstacle or unexpected event in front of the vehicle, it is necessary to locate and predict the trajectory of both the front target and the vehicle motion state [3]. Motion target positioning prediction can provide a practical basis for the application of autonomous collaborative driving, collision warning, anti-collision, and other related safety technologies in intelligent transportation systems, such as detection and tracking of motion targets in the process of motion target path prediction, and the prediction of motion target positioning on this basis. This is in order to visually alert the driver and issue warnings or take active control of vehicle braking, steering, and other measures to avoid traffic accidents. Therefore, motion target location prediction research is a key factor to accelerate the application of intelligent transportation systems. At present, the commonly used methods include traditional methods, machine learning-based methods, and deep learning-based methods.

The traditional method is mainly based on the assumption of object kinematics, and the prediction results are calculated by establishing a kinematic or kinetic model to estimate the propagation of the object motion state over time. Kim et al. [4] defined the desired yaw rate required for lane changes and curves and added the desired yaw rate to the extended Kalman filter. To further improve the prediction accuracy, Schreier et al. [5] proposed a probabilistic trajectory prediction method for Monte Carlo simulation. However, this method cannot accurately capture the complex and variable motion characteristics of dynamic objects, can only obtain accurate results in a very short period of time, and does not match the real trajectory in predictions of more than 1 s, which may not be effective in practical applications. The main machine learning-based methods for trajectory prediction are Gaussian processes, Hidden Markov Models, and Bayesian networks. Streubel et al. [6] used Hidden Markov models to predict discrete actions of each object independently. However, the assumption of total independence usually does not hold in real scenarios. To consider more complex models of vehicle interactions, Gindele et al. [7] used a dynamic Bayesian network to predict the trajectories of vehicles, but the network is computationally more expensive. Moreover, these methods are not simple for handling high-dimensional data and require hand-designed input features to capture contextual information, all of which limit the flexibility of the learning algorithms and thus lead to poor performance. In addition, these methods can only predict the behavior of specific entities. With the success of deep learning in various fields of computer vision and robotics, a number of researchers have started to introduce deep learning into trajectory prediction tasks. Nan Li [8] used a reverse neural network to establish a flight trajectory prediction model that can predict the multidimensional characteristics of flight trajectory; however, for pedestrian trajectory prediction in a long- and short-term memory network, poor real-time prediction accuracy is insufficient. The flight trajectory prediction method based on a convolutional neural network proposed by Hongpeng Zhang [9] compares with a long- and short-term memory network and a fully connected network whose operation speed and error are smaller, and which also has a good ability to learn and adapt to the dynamic characteristics of an uncertain system, although the trajectory prediction value obtained for ping pong is poor due to the influence of rotation state and bounce force. However, this method is all about separating spatial information from temporal information in dynamic object trajectory localization prediction.

Due to the complexity of urban scenes, where the future motions of pedestrians and cars are influenced by the motions of other objects and their spatial environment, to improve the accuracy of trajectory prediction researchers have started to consider modeling the social interactions between multiple objects and modeling the constraints of scene context for object trajectory prediction. Alahi et al. [10] proposed a Social-LSTM model to capture the social interactions of pedestrians by running maximum pooling on the state vectors of nearby pedestrians over a predefined range of distances, but such interactions were not modeled for social interactions of distant pedestrians. Sadaghian et al. [11] proposed an attentional recurrent neural network with inputs of pedestrian past motion trajectories and top-view images of navigation scenes to obtain more accurate prediction results by learning the influence of the spatial environment on pedestrian trajectories. The trajectory prediction algorithm based on ARIMA-UKF proposed by Nanhua Chen et al. [12] achieves the prediction estimation of the target by first using a traceless Kalman filter in the case of scalar maneuver change, then predicting the target acceleration information using the autoregressive integrated sliding average model, and finally by combining the traceless Kalman filter again to achieve the prediction, which can also produce the accuracy prediction for the nonlinear and finite-dimensional linear spatial trajectory results. Although there are some research results on predicting pedestrian trajectories in complex urban scenes, there are fewer papers on predicting vehicles or trajectories of multiple types of objects. Moreover, in most of the studies, the spatial information is separated from the temporal information. The temporal information is used as a known and determined parameter to predict the target motion trajectory by the change in the target’s position in space in the sequence image, combined with the known time interval information.

In summary, this paper proposes to establish a time-space four-dimensional joint imaging model, based on the existing research, for the application of machine vision and to explore the problem of motion target localization and prediction based on this model. Specifically, it is proposed to extend the optical imaging model—based on the spatial object-image relationship, itself based on the principle of machine vision central perspective projection—to include the temporal dimension to establish a time-space four-dimensional joint imaging model; to build a motion trend learning sample library of motion targets based on artificial neural network technology in order to learn the object-image mapping relationship; and to predict the motion localization of motion targets.

## 2. Methodology

### 2.1. Time-Space Four-Dimensional Joint Imaging Model

In this paper, we propose a schematic diagram of the time-space four-dimensional joint imaging model as shown in Figure 1. Based on the central perspective imaging model, the spatial object-image relationship is expanded along the time axis and into a time-space four-dimensional imaging system to establish a visual imaging system model that incorporates the time dimension, that is, the time-space four-dimensional joint imaging model.

In this model, the camera coordinate system and image coordinate system are established. The origin of the camera coordinate system is taken as the camera optical center O, the ${\;Z}_{C}$-axis coincides with the camera optical axis, and the camera direction is taken as positive. The image coordinate system is established in the ortho-image plane $S^{\prime }$, which lies in the plane of the camera coordinate system, and f  is the central perspective projection ${\;Z}_{C}$ = f focal length. The image physical coordinate system is established in the image plane $S^{\prime }$. The origin of the image physical coordinate system is the intersection o of the camera optical center O and the image plane $S^{\prime }$. The x-axis and y-axis of the image physical coordinate system are parallel to the ${\;X}_{C}$-axis and ${\;Y}_{C}$-axis of the camera coordinate system, respectively. The image pixel coordinate system I-uv is established, with the point I in the upper left corner of the image as the origin and the pixel as the coordinate unit. uv denotes the number of columns and rows of the pixel in the digital image, respectively, consistent with the common storage format of digital image pixels. ${\;t}_{0}$, ${\;t}_{1}$, ${\;t}_{2}$ are the moments corresponding to the sequence image [13]. Take uniform linear motion as an example; the camera takes pictures with a certain frame rate N exposure. Let the moving target be in uniform linear motion in the object plane S perpendicular to the optical axis of the camera, and the digital image taken at moment ${\;t}_{0}$ is ${\;F}_{0}$. At moment $t_{1}$($t_{1}=t_{0}+1/N$), the target point is at position $P_{t1}$ on the object plane S; at moment $t_{2}$($t_{2}=t_{1}+1/N=t_{0}+2/N$), the target point is at position $P_{t2}$ on the object plane S; at moment $t_{3}$($t_{3}=t_{2}+1/N=t_{1}+2/N$), the target point is at position $P_{t3}$ on the object plane S; at moment $t_{n}$($t_{n}=t_{n-1}+1/N=t_{n-2}+2/N$), the target point is at position $P_{m}$ on the object plane S.

In the central perspective projection imaging model, the relationship between a point in the world coordinate system space and the image point it images in the image plane is shown in Formula (1). The inner reference matrix of the imaging model, which contains parameters such as principal points and equivalent focal lengths, describes the characteristics of the camera itself [14]; the outer reference matrix, which is the rotation matrix and translation vector of spatial points in the world coordinate system concerning the camera coordinate system, describes the relative position and pose relationship between the camera coordinate system and the world coordinate system [15].
(1)$$\left[\begin{array}{c} u\\ v\\ 1 \end{array}\right]=\left[M_{\mathrm{inside}}\right]\times \left[M_{\mathrm{outside}}\right]\left[\begin{array}{c} X\\ T\\ Z\\ 1 \end{array}\right]$$
(2)$$\left[M_{\mathrm{inside}}\right]=\left[\begin{array}{c} F_{u}\;\;\;\;\;0\;\;\;\;\;\;C_{u\;}\;\;\;0\\ \;\;\;0\;\;\;\;\;\;\;\;F_{v}\;\;\;\;C_{v}\;\;\;\;0\;\;\\ 0\;\;\;\;\;\;\;0\;\;\;\;\;\;\;\;1\;\;\;\;\;0 \end{array}\right]$$
(3)$$\left[M_{\mathrm{outside}}\right]=\left[\begin{array}{c} R\;\;\;\;T\\ 0^{T}\;\;\;1 \end{array}\right]$$

In Formula (2), ($C_{u},C_{v})$ is the digital image coordinate of the digital image principal point, which is the intersection of the optical axis and the image plane o; ($F_{u},{\;F}_{v})$ is the equivalent focal length, $F_{u}=f/d_{x}$, $F_{v}=f/d_{y}$, f is the camera focal length. In Formula (3), R is the rotation matrix; it is a unit orthogonal array of 3×3. Its 9 elements are a combination of trigonometric functions of the rotation angle ($A_{x},A_{y},A_{z})$, which is defined as the Eulerian angle that transforms the world coordinate system to the same attitude as the camera coordinate system and turns around each of the three axes; $T=\left(T_{x},T_{y},T_{z}\right)$ is the translation vector, which is the coordinate of the origin of the world coordinate system in the camera coordinate system.

Following the idea that the internal and external reference matrices of the camera calibration under the spatial object-image model are set up independently, the time parameters and the target motion parameters are proposed independently to establish the object-image mapping relationship based on the time-space model in Figure 1 above. As shown in Formula (4), the relationship between a point P(X,Y,Z) in the time-space of the world coordinate system and its image point P(u,v) is:(4)$$\left[\begin{array}{c} u\\ v\\ 1 \end{array}\right]=\left[M_{\mathrm{inside}}\right]\cdot \left[M_{\mathrm{outside}}\right]\cdot \left[M_{t}\right]\cdot \left[M_{v}\right]\cdot \left[\begin{array}{c} X\\ T\\ Z\\ 1 \end{array}\right]$$

In Formula (4), $\left[M_{t}\right]$ is the hardware time performance; $\left[M_{v}\right]$ is the motion parameter matrix and consists of parameters describing the operational state of the target object, which may include parameters such as the speed of motion.

Taking uniform linear motion as an example, the establishing of object and image point mapping relationships in a joint time-space four-dimensional imaging model is performed. When time and motion relationships are not considered, the coordinate calculation relationships shown in Formulae (5) and (6) are satisfied for independent object and image point pairs under the central perspective imaging model.
(5)$$Z_{c}\left[\begin{array}{c} u_{\mathrm{pt}0}\\ v_{\mathrm{pt}0}\\ 1 \end{array}\right]=\left[\begin{array}{c} F_{u}\;\;\;\;0\;\;\;\;\;C_{u\;}\;\;\;0\\ \;\;\;0\;\;\;\;\;\;F_{v}\;\;\;\;C_{v}\;\;\;0\;\;\\ 0\;\;\;\;\;\;\;0\;\;\;\;\;\;\;1\;\;\;\;\;0 \end{array}\right]\left[\begin{array}{c} X_{c-\mathrm{pt}0}\\ Y_{c-\mathrm{pt}0}\\ Z_{c-\mathrm{pt}0}\\ 1 \end{array}\right]$$
(6)$$\left[\begin{array}{c} X_{c-\mathrm{pt}0}\\ Y_{c-\mathrm{pt}0}\\ Z_{c-\mathrm{pt}0}\\ 1 \end{array}\right]=\left[\begin{array}{c} R\;\;\;\;T\\ 0^{T}\;\;\;1 \end{array}\right]\left[\begin{array}{c} X_{\mathrm{pt}0}\\ Y_{\mathrm{pt}0}\\ Z_{\mathrm{pt}0}\\ 1 \end{array}\right]$$

In Formulae (5) and (6), ($X_{c-\mathrm{pt}0},Y_{c-\mathrm{pt}0},Z_{c-\mathrm{pt}0})$ are the coordinates of the object point $P_{t0}$ in the camera coordinate system O-($X_{c},Y_{c},Z_{c})$, ($u_{\mathrm{pt}0},v_{\mathrm{pt}0})$ is the coordinate of the corresponding image point $P_{t0}$ in the physical coordinate system O-xy of the image, and ($X_{\mathrm{pt}0},Y_{\mathrm{pt}0},Z_{\mathrm{pt}0})$ is its coordinate at the object point $P_{t0}$ in the world coordinate system W-XYZ.

After considering the time and motion relationship, let the point $P_{0}$ be the location of the object point at the initial moment of camera exposure; under the camera coordinate system, let its coordinates be ($X_{C-p0},Y_{C-p0},Z_{C-p0})$. In the exposure cycle with serial number n (the nth image of the sequence image), the object point is located at position $P_{\mathrm{tn}}$ and its coordinates are ($X_{C-\mathrm{ptn}},Y_{C-\mathrm{ptn}},Z_{C-\mathrm{ptn}})$ under the camera coordinate system. Under the operator linear motion model, $P_{\mathrm{tn}}$ has the following coordinate relationship with the initial point $P_{t0}$.
(7)$$\left\{ \begin{array}{l} X_{C-ptn}=X_{C-p0}+(t_{e-d}+t_{e-c})\times V_{xc}+n\times t_{frame}\times V_{xc}\\ Y_{C-ptn}=Y_{C-p0}+(t_{e-d}+t_{e-c})\times V_{yc}+n\times t_{frame}\times V_{yc}\\ Z_{C-ptn}=Z_{C-p0}+(t_{e-d}+t_{e-c})\times V_{zc}+n\times t_{frame}\times V_{zc} \end{array}\right.$$

The time parameters in Formula (7) are extracted independently from the motion state parameters and written in the form of matrix multiplication.
(8)$$\left[\begin{array}{c} X_{C-ptn}\\ Y_{C-ptn}\\ Z_{C-ptn}\\ 1 \end{array}\right]=\left[\begin{array}{cccc} V_{xc} & 0 & 0 & X_{C-p0}\\ 0 & V_{yc} & 0 & Y_{C-p0}\\ 0 & 0 & V_{zc} & Z_{C-p0}\\ 0 & 0 & 0 & 1 \end{array}\right]\cdot \left[\begin{array}{cccc} n & 1 & 1 & 0\\ n & 1 & 1 & 0\\ n & 1 & 1 & 0\\ 0 & 0 & 0 & 1 \end{array}\right]\cdot \left[\begin{array}{c} t_{frame}\\ t_{e-d}\\ t_{e-c}\\ 1 \end{array}\right]$$

In Formula (8), $t_{\mathrm{frame}},{\;t}_{e-d},t_{e-c}$ indicates the camera exposure clock period, exposure delay, and exposure duration. The matrix composed of ($t_{\mathrm{frame}},t_{e-d},t_{e-c})$ is the time parameter matrix, reflecting several important time parameters of the camera; the matrix composed of ($V_{\mathrm{xc}},V_{\mathrm{yc}},V_{\mathrm{zc}})$ is the motion state matrix, reflecting the motion state and initial state of the target; the matrix containing the parameter n is mainly used to determine the position of the analyzed image in the sequence image and the time point at which the current target point is located.

Substitute Formula (8) into Formula (5) to obtain Formula (9). Then, using the idea of conversion between coordinate systems shown in Formula (6), it obtains Formula (4).
(9)$$Z_{C}\left[\begin{array}{c} u_{ptn}\\ v_{ptn}\\ 1 \end{array}\right]=\left[\begin{array}{cccc} F_{u} & 0 & C_{u} & 0\\ 0 & F_{v} & C_{v} & 0\\ 0 & 0 & 1 & 0 \end{array}\right]\cdot \left[\begin{array}{cccc} V_{xc} & 0 & 0 & X_{C-p0}\\ 0 & V_{yc} & 0 & Y_{C-p0}\\ 0 & 0 & V_{zc} & Z_{C-p0}\\ 0 & 0 & 0 & 1 \end{array}\right]\cdot \left[\begin{array}{cccc} n & 1 & 1 & 0\\ n & 1 & 1 & 0\\ n & 1 & 1 & 0\\ 0 & 0 & 0 & 1 \end{array}\right]\cdot \left[\begin{array}{c} t_{frame}\\ t_{e-d}\\ t_{e-c}\\ 1 \end{array}\right]$$

The advantages of independently proposing the internal reference matrix, external reference matrix, temporal parameter matrix, and target motion parameter matrix to establish a joint time-space four-dimensional imaging model are: the parameters are connected with each other and independent of each other; and the four-dimensional time-space localization of sequential images under motion targets incorporates the existing multi-directional correlation techniques and provides an important means for high-level dynamic scene structure localization, which is an important part of realizing the upgrade of existing target localization techniques.

When extending the spatial imaging model to a spatio-temporal four-dimensional imaging model, a problem that must be faced is the complication of the system computation due to the increase in the system parameters and its solution. To avoid matrix singularities, the constraint relations of the imaging system need to be increased; when certain cost function minimization algorithms are used, the large number of parameters will lead to poor stability of the algorithm, and the algorithm may converge to the wrong solution, or even lead to algorithm failure. Artificial neural network technology, for complex systems of computational problems, provides a solution to the problem [16]. The artificial neural network is a huge, interconnected network; each neuron has a relatively simple structure without a fast processing speed, but the network has a more than three orders of magnitude higher number of connections than neurons. This architecture theory of connectionism, both for knowledge representation and the information processing process, presents a new idea [17]. The artificial neural network allows constituting a self-learning and adaptive architecture in the interaction with external information to form a nonlinear mapping or nonlinear dynamical system to correctly reflect the relationship between input and output without knowing the precise mathematical model of this relationship in advance, and to establish the object-image mapping relationship in a new way [18].

### 2.2. Prediction Model

The main idea of the backpropagation algorithm in BP neural networks is to divide the learning process into two parts: in the first stage (forward propagation process), the input information is calculated from the input layer to obtain the actual output value of each unit layer, and the state of each layer of neurons only affects the state of the neurons in the next layer [19]; in the second stage (backpropagation process), assuming that the desired output value is not obtained in the output layer, the difference between the actual output and the desired output is calculated recursively layer by layer, and the error signal is often modified by modifying the previous weight minimization layer according to the error [20]. It calculates the direction of decline relative to the slope of the error function by continuously calculating the direction of the change in the network weights and deviations. The change in each weight is proportional to the error.

The specific operation process of the BP neural network is described below.

(1)Initialization. The weight matrices W and V are assigned random numbers, the error E is set to 0, the learning rate η is set to a small number in the (0–1) interval, and the accuracy $E_{\mathrm{MIN}}$ achieved by the network after training is set to a positive small number.

(2)Forward input. The training samples are input and the output of each layer is calculated in turn, where f(·) is the selected neuron activation function.


(10)
$$Y=f\left(V^{T}X\right)$$



(11)
$$O=f\left(W^{T}Y\right)$$


(3)Calculate the network output error. The root mean square error $E_{\mathrm{RMSE}}$ is used as the total output error of the network, where P is the total number of samples.


(12)
E= d−O



(13)
$$E_{\mathrm{RMSE}}=\sqrt{\frac{1}{P}\sum_{P=1}^{P}E^{2}}$$


(4)Back propagation of the error signal. The backward output error is calculated layer by layer for each neuron in each layer. Here, the activation function is sigmoid. In the following equation, k is the output layer neuron label, j is the hidden layer neuron label, and l is the number of neurons in the output layer.


(14)
$$\delta_{k}^{o}=\left(d_{k}-o_{k}\right)o_{k}(1-{\;o}_{k})$$



(15)
$$\delta_{j}^{y}=(\sum_{k=1}^{1}\delta_{k}^{o}\omega_{\mathrm{jk}})y_{j}(1-{\;y}_{j})$$


(5)Weight adjustment. The weight adjustment component $\omega_{\mathrm{jk}}$ between the *j*th hidden layer neuron and the kth output layer neuron, and the weight adjustment component $\Delta v_{\mathrm{ij}}$ between the jth hidden layer neuron and the i-th input layer neuron are shown in the following Formulae.


(16)
$$\Delta \omega_{\mathrm{jk}}={\;\eta \delta }_{k}^{o}y_{j}=\eta \left(d_{k}-o_{k}\right)o_{k}\left(1-o_{k}\right)y_{j}$$



(17)
$$\Delta v_{\mathrm{ij}}={\;\eta \delta }_{j}^{y}x_{i}=\eta \left(\overset{l}{\underset{k=1}{\sum }}\delta_{k}^{o}\omega_{\mathrm{jk}}\right)y_{j}\left(1-y_{j}\right)x_{i}$$


(6)The adjusted weights are used to recalculate the output error from the input layer and determine whether the total error of the network meets the accuracy requirement. For example, if $E_{\mathrm{RMSE}}$ is used as the total error of the network, if $E_{\mathrm{RMSE}}$ < ${\;E}_{\mathrm{MIN}}$, the training is finished; otherwise, the above steps (10) to (17) are repeated.

The purpose of the BP neural network-based prediction model in this paper is to establish the mapping relationship between the two-dimensional pixel coordinates of digital images and the time-space four-dimensional coordinates. The binocular vision structure is used, and the image point corresponding to an object point in the real world consists of a left-view image point and a right-view image point. Therefore, the input layer of the neural network is set to 4 nodes, which correspond to the image pixel coordinates in the left and right views of the object (ul,vl,ur,vr); the output layer is set to 4 nodes, which correspond to the time-space four-dimensional coordinates of the object in the real world; the origin of the world coordinate system is the location of the optical center of the left camera.

So far, there is no clear theory on how to determine the number of neurons in the hidden layer, which is usually determined using empirical formulae and multiple trial methods. One theory now suggests that the number of neurons in the hidden layer is equal to two times the number of neurons in the input layer plus one. The preliminary experiments in this paper use a three-layer neural network structure, that is, an input layer (4 input nodes), an output layer (4 output nodes), and a hidden layer with 10 nodes set in the hidden layer, which is a 4-10-4 type BP network, as shown in Figure 2.

In this paper, the neural network uses the logarithmic function as the activation function. The initial weights are generally chosen as random numbers between (−1,1). Since the system is nonlinear, the initial value plays an important role in whether the learning reaches a local minimum, whether it can converge, and how long the training time is. If the initial value is too large, it makes the weighted input fall into the saturation region of the activation function, which leads to a very small derivative, and, in the calculation of the weight correction formula, when converging to 0 it makes the weights converge to 0, which makes the regulation process almost stop, so it is desired that the output value of each neuron after the initial weighting is close to 0 in order that the weights of each neuron can be guaranteed. The adjustment is performed at the point where their activation function changes the most.

The learning rate determines the amount of weight change generated in each training cycle. A large learning rate may lead to instability of the system, while a small learning rate will result in a long training time and slow convergence but ensures that the error value of the network will not jump out of the trough of the error surface and eventually converge to the minimum error value. Therefore, the learning rate of 0.1 was chosen in this paper.

In summary, the neural network structure and parameters used in the experiments of this paper are shown in Table 1.

### 2.3. Forecasting Process

This paper is based on a machine vision approach to learn and predict the motion trajectory of a moving target in the spatial and temporal regime. Therefore, the basic framework of this paper uses parallel binocular vision structure. The binocular stereo vision measurement method has the advantages of high efficiency, suitable accuracy, simple system structure, and low cost. It is one of the key technologies of computer vision. Obtaining distance information of a spatial three-dimensional scene is also the most basic content in computer vision research [21].

In this paper, we propose to use two high-speed cameras (see Section 3.1 for model details) in the laboratory to form a binocular stereo vision system for motion target measurement and continuously acquire sequential images of motion targets with a certain temporal resolution. The binocular cameras are mounted parallel to the optical axis, and a certain base distance is maintained between the two cameras to ensure that the binocular camera field of view covers the target motion range.

The parallel binocular vision structure, the 3D world coordinates of the target point, can be calculated by the parallax method. The parallax D of the spatial point P in the two image planes is defined as the difference in pixel coordinates in the x-direction, as shown in Formula (18), where $u_{l}$ and $v_{r}$ denote the pixel coordinates of the target point in the column direction in the left and right views, respectively; the 3D world coordinates of the target point can be calculated according to the triangle principle (the origin of the world coordinates is the location of the left camera optical center), as shown in Formula (19), where $u_{l}$ and $v_{l}$ denote the pixel coordinates of the target point in the left view in the column and row direction, $u_{0l}$ and $v_{0l}$ denote the pixel coordinates of the left camera optical center in the left view, B is the base distance between binocular cameras, and f is the equivalent focal length of the camera. $u_{0l}$, $v_{0l}$, B, and F can be obtained by camera calibration.
(18)$$D=u_{l}-u_{r}$$
(19)$$\{\begin{array}{c} x=\frac{B\;\cdot \;u_{l}}{D}\\ Y=\frac{B\;\cdot \;\mathrm{vl}}{D}\\ Z=\frac{B\;\cdot \;F}{D} \end{array}$$

In summary, the prediction process adopted in this paper is shown in Figure 3. Firstly, this experiment was based on a parallel binocular stereo vision structure; therefore, an experimental system consisting of a binocular camera, a guide rail, and a checkerboard grid target was constructed first. Secondly, a data sample library was constructed to record the time-space data of the corner points of the checkerboard grid at different locations at different moments by using the guide rails to drive the checkerboard grid targets to move at different speeds for the output and input of the neural network. Thirdly, we trained neural networks using a calibrated sample pool, binning training and test samples by random method, verifying the learning and prediction performance on time-space four-dimensional data, and performing error analysis. Finally, a binocular camera was used to capture the left and right views of the moving object, and the trained network was used to predict the time-space localization of the object. The final realization was to be able to learn the object-image mapping relationship model in four dimensions of time-space using the BP neural network.

## 3. Experiments and Results

### 3.1. Expeimental Equipment

In order to verify the feasibility of the model and the performance of the proposed calibration method, a series of experiments were conducted. An experimental system consisting of a binocular camera, a guide rail, and a checkerboard grid target was built as shown in Figure 4b. Two high-speed cameras of the same type were mounted horizontally parallel to the optical axis, and a guide rail was arbitrarily placed in front of the camera to drive the movement of the checkerboard target on the guide rail through the control of stepper motors. When the calibration board was used for system calibration, only the binocular camera could be triggered for image acquisition of the calibration board. In the experimental process of this paper, two high-speed cameras were selected to shoot the moving objects in real time. The high-speed camera is shown in Figure 4a. The hardware equipment used in this experiment is shown in Table 2.

### 3.2. System Calibration

The imaging system was calibrated using the Zhengyou Zhang calibration method. A homemade laboratory tessellation grid calibration board with a tessellation grid size of 20 mm × 20 mm and 128 arrays of corner points was used. The multi-directional checkerboard grid targets were acquired by a binocular camera, and the imaging system calibration parameters were resolved based on the corner point data of the acquired multi-directional images. Figure 5 and Figure 6 show the system calibration with the left camera and the right camera, respectively; Table 3 shows the calibration results of the system parameters in this paper.

### 3.3. Sample Bank Construction

The screw guide is placed along the direction of the optical axis of the binocular camera (with the target running, the depth z distance is changed), the target plate is carried by the guide table running at a certain speed, and the coordinate value of each spatial position of each corner point of the target plate at each moment is recorded, which constitutes a time-space four-dimensional data sample library. For each corner point, the pixel coordinates (ul,vl,ur,vr) of its left and right views are the input values corresponding to the four nodes in the input layer of the neural network. In accordance with the binocular stereo vision theory and system calibration results, the world 3D coordinates (x,y,z), corresponding to each corner point, and the frame image sequence number (corresponding to the moment value t) are obtained, which are the output values corresponding to the four nodes in the output layer of the neural network. Due to the limited length of the guide, the number of frames of the intercepted image sequence will be more when the operation is slow and less when the operation is fast. Figure 7 and Figure 8 show examples of image acquisition by the left and right cameras, respectively, when the guide carries the target movement during the creation of the sample library.

Group 1: We set the guide rail to carry the target to perform uniform linear motion with speed v = 20 mm/s; we set the camera to continuously shoot the image of the moving target plate with frame rate 50 fps; the target moved from the starting point of the guide rail to the end of the guide rail, and a total of 601 frames of video images were obtained. The images with frame numbers 30, 60, 90, 120, 150, 180, 210, 240, 270, 300, 330, 360, 390, 420, 450, 480, 510, 540, and 570 were extracted from the video, and there were 12819 = 1824 corner points. When used for neural network training, 90% of the points are randomly selected as training samples and the remaining 10% are used as test samples, which means the number of training samples was 1642 data and the number of test samples was 182 data.

Group 2: We set the guide rail to carry the target to perform uniform linear motion with speed v = 40 mm/s; we set the camera to continuously shoot the image of the moving target plate with frame rate 50 fps; the target moved from the beginning of the guide rail to the end of the guide rail, and a total of 310 frames of video images were obtained. The images with frame numbers 15, 30, 45, 60, 75, 90, 105, 120, 135, 150, 165, 180, 195, 210, and 225 were extracted from the video, and there were 12,815 = 1440 corner points. When used for neural network training, 90% of the points were randomly selected as training samples and the remaining 10% were used as test samples, which means the number of training samples was 1296 data and the number of test samples was 144 data.

Group 3: We set the guide rail to carry the target to perform uniform linear motion with speed v = 60 mm/s; we set the camera to continuously shoot the image of the moving target plate with frame rate 50 fps; the target moved from the beginning of the guide rail to the end of the guide rail, and a total of 220 frames of video images were obtained. The images with frame numbers 15, 30, 45, 60, 75, 90, 105, 120, 135, 150, 165, 180, and 195 were extracted from the video, and there were 12813 = 1248 corner points. For neural network training, 90% of the points were randomly selected as training samples and the remaining 10% were used as test samples, which means the number of training samples was 1123 data and the number of test samples was 125 data.

### 3.4. Results

#### 3.4.1. Experiment on Time-Space Positioning Prediction of Corner Point Sample Banks

A sample bank consisting of three sets of target corner points with different velocities was used for the motion localization prediction experiments in the previous Section 3.3. For each group of sample banks, 90% of its sample points were randomly selected for artificial neural network training; after the training was stabilized, the remaining 10% of its sample points were used for testing to evaluate the test points in the time-space four-dimensional localization prediction. Figure 9, Figure 10 and Figure 11 show the time-space localization prediction results and the absolute error results of the four-dimensional prediction for the test sample points of groups 1, 2, and 3 of the sample banks, respectively.

To visualize the time-space localization of the test sample points, a three-dimensional coordinate system combined with color is used to display the four-dimensional information, as shown in Figure 9a, Figure 10a and Figure 11a. The 3D coordinate positioning shows the spatial positioning, and the color values mapped on the color axis show the temporal positioning, where the data1 hollow circle “°” indicates the predicted positioning value of the neural network for the test sample points, and the data2 asterisk “∗” indicates the true value of the test sample points. 

Figure 9b, Figure 10b and Figure 11b show the four-dimensional absolute error distributions of the predicted values of the test points, which are the absolute error values of the four-dimensional data using the formula shown in Formula 20 below, where ${\hat{x}}_{i},{\hat{y}}_{i},{\hat{z}}_{i},{\hat{t}}_{i}$ and $x_{i},y_{i},z_{i},t_{i}$ denote the predicted and measured values of the four-dimensional coordinates, respectively.
(20)$${\mathrm{error}}_{i}=\sqrt{\frac{{\left({\hat{x}}_{i}-x_{i}\right)}^{2}+{\left({\hat{y}}_{i}-y_{i}\right)}^{2}+{\left({\hat{z}}_{i}-z_{i}\right)}^{2}+{\left({\hat{t}}_{i}-t_{i}\right)}^{2}}{4}}$$

From Figure 9, Figure 10 and Figure 11, it can be seen that the hollow circles basically overlap with the asterisk positions, regardless of the velocity values, and the predicted color mapping values do not differ greatly, i.e., the four-dimensional coordinates tested by the network basically match the true value data. In terms of prediction errors, the network prediction error data (average absolute error data) in the range of x and y coordinates are hundreds of millimeters, z coordinates are thousands of millimeters, and t coordinates are units of seconds. According to the positioning absolute error value calculated by Equation (18), distribution in 0–5.0 is concentrated below 2.5, and there are very few error data greater than 2.5. Therefore, it has been proven that, not only can the network establish the corresponding spatio-temporal object mapping relationship between the motion target and the sequence image, but also that the prediction error is relatively small. Therefore, it has been proven that the model can better achieve the prediction of spatio-temporal four-dimensional localization of moving targets.

#### 3.4.2. Experiment on Time-Space Localization Prediction of Real Object Motion

Based on the experiments in Section 3.4.1 above, which predicted the motion of three sets of checkerboard targets with different speeds, this section further investigates replacing the checkerboard targets with real objects, placing them horizontally on the table on the guide rail and making them move forward in the direction of the camera. The camera acquisition software was used to record the object running at a certain speed. The pixel coordinates of the left and right views of the center of mass of the object at each position were obtained by image processing and fed into the neural network trained with the target sample points (in Section 3.4.1) in order to obtain the time–space localization data of the object output from the network and to allow evaluation of the time-space localization error. 

Matching the velocities of the target samples in Section 3.4.1, the images of the physical motion sequences of 20 mm/s, 40 mm/s, and 60 mm/s were recorded using the binocular camera acquisition software. Figure 12 and Figure 13 show examples of the images captured by the left and right cameras, respectively, when the guide translating stage carries the physical object in the physical motion prediction experiment.

As a basic research area in the field of computer vision, object recognition tries to identify what objects exist in images and to report the position and orientation of these objects in the images [22]. After the image acquisition is completed, the coordinates of the center-of-mass points of the target object are extracted from each frame of each group of videos. The flowchart of the image processing algorithm is shown in Figure 14—including grayscale map binarization, threshold selection, morphological processing, edge extraction, skeleton processing, etc.—to obtain the bounding box of the target object as well as the left and right view pixel coordinate values of the target center-of-mass x. The extracted bounding boxes and center-of-mass points of the target object are labeled as shown in Figure 14 and Figure 15.

Three groups of physical objects with different velocities moved from the beginning of the guide rail to the end of the guide rail, and a total of 501, 301, and 201 frames of video images were obtained for each group (the length of the guide rail is fixed, and the number of video frames obtained is different for different movement speeds). The data of 80 sets of mass points of the moving target are randomly extracted from each of the three groups of videos and input into the neural network model trained with the corresponding velocity corner point sample library in the aforementioned experimental session for the prediction of physical positioning. The four-dimensional images of the output data are presented in Figure 16 below. Figure 16a shows the motion trajectory of the object when the speed is 20 mm/s, Figure 16b shows the motion trajectory of the object when the speed is 40 mm/s, and Figure 16c shows the motion trajectory of the object when the speed is 60 mm/s. In these figures, the output predicted by the neural network is represented by the data1 hollow circle “°”; the 3D world coordinates obtained by the traditional binocular parallax algorithm are represented by the data2 asterisk “∗”; and the color mapping represents the relationship of the time change: as the time increases, the graph color also changes according to the color mapping axis.

As shown in Figure 16, the target points at different velocities, fed into the trained corresponding neural network model, can be well predicted for temporal and spatial localization. Spatially, the target moves toward the camera along the camera optical axis direction (z-axis direction), and Figure 17 can accurately reflect the spatial localization development curve as a straight line along the z-axis direction. Temporally, the continuous change in time is reflected by the color mapping; the process of time change reflected by the color development change can also be accurately predicted, and the predicted time value matches very well with the real-time value collected by the high-speed camera.

Figure 17 shows the depth z values predicted by the neural network compared with the depth z values obtained by the conventional binocular parallax algorithm for three different speeds in the experiments. The gray curve is the depth value obtained by the binocular parallax algorithm and the red curve is the depth value predicted by the neural network output. Figure 18 shows the error curves of the corresponding points of the two curves in each subplot of Figure 17. As shown in the figure, the absolute error of the depth information prediction value for the speed group of 20 mm/s is roughly distributed between −2.06 and 2.23, and the relative error rate is distributed between −0.18% and 0.19%; the error of the depth information prediction value for the speed group of 40 mm/s is roughly distributed between −1.94 and 2.09, and the relative error rate distribution is −0.17~0.18%; the error of the depth information prediction value for the speed group of 60 mm/s is roughly distributed between −2.11 and 2.76, and the relative error rate distribution is −0.19~0.23%. Regarding the slight change in error floating, when the guideway is running slower or faster, the motion stability is slightly worse and the error of the target depth prediction is slightly larger; when the motion speed is uniform, the error is the smallest and the consistency in prediction is the best.

Figure 19 shows the time values predicted by the network for three different speeds compared with the standard time values, which are calculated from the frame difference values of the image sequences and the exposure frequency values of the high-speed camera. The gray curve is the time standard value and the red curve is the time value of the predicted output of the neural network. Figure 20 shows the error curves of the corresponding points of the two curves in each subplot of Figure 19. As shown in the figure, the time error at a speed of 20 mm/s is roughly distributed between −0.12 and 0.06, with a relative error rate distribution of −1.51%~1.02%; the time error at a speed of 40 mm/s is roughly distributed between −0.09 and 0.08, with a relative error rate distribution of −2.01%~1.23%; the time error at a speed of 40 mm/s is roughly distributed between −0.09 and 0.08, with a relative error rate distribution of −2.01%~1.23%; and when the speed is 60 mm/s, the time error is roughly distributed between −0.08 and 0.07, and the relative error rate is −1.82%~2.03%.

Finally, the predicted values of the target running speeds are compared and analyzed in this paper to verify the accuracy of the time-space four-dimensional data values predicted by the neural network. The time-space data values predicted by the neural network were used to calculate the running speed of the physical target as the speed prediction value; the values of the guideway moving speed set during the experiment were used as the real value of speed for the error analysis of speed prediction. From the three sets of experiments with different speeds, six sets of data were arbitrarily extracted for tabular presentation, as shown in Table 4 where (a), (b), and (c) represent the comparison between the predicted velocity results and the standard velocity values when the velocity was 20 mm/s, 40 mm/s, and 60 mm/s, respectively. From the data in Table 4, it can be seen that the network model can predict the velocity corresponding to each moment of the target point under each velocity sample better and with less error. Therefore, this experiment shows that the time-space four-dimensional model proposed in this paper, trained by sample points, can well learn the mapping relationship from image-space pixel coordinates to time-space four-dimensional coordinates in binocular vision, and can still achieve the time–space localization prediction from image point to object point even if the samples have different speed differences.

#### 3.4.3. Comparisons with Other Works

To further demonstrate the feasibility and advantages of the model in this paper, the advantages and disadvantages of the spatio-temporal four-dimensional target localization prediction model were compared with various prediction model algorithms in [23,24,25,26,27], as shown in Table 5.

The Kalman filter (KF) can estimate the state behavior of a moving target while predicting the trajectory at the next moment using new observations. The authors of [23] applied KF to vehicle trajectory prediction over short distances and obtained good prediction results after validation with real moving target datasets. However, the algorithm is poor in real time and cannot predict the trajectory of long-time movement. With the continuous development of artificial neural networks, their related techniques have been applied to trajectory prediction. In some simple prediction tasks, the BP neural network model can input the historical trajectory data of the moving target into the neural network for training in order to learn the motion characteristics of the target trajectory, and then predict the future trajectory based on those motion characteristics. The authors of [24] proposed a BP neural network-based airborne target trajectory prediction model with strong robustness for the target trajectory prediction problem in hotspot areas. However, the output is the target rule trajectory, which does not reflect the real-time nature of the trajectory. The differential autoregressive moving average model predicts future trajectories by linearly combining a large amount of historical trajectory data and current trajectory data. The authors of [25] added feature data to the historical trajectory data for prediction, which significantly improved the model prediction effect. However, the ARIMA model is widely used for univariate trajectory prediction, and the model only calculates the average value of the nearest position point; moreover, it is difficult to capture the motion target in practical application, which has some limitations. RNN can capture temporal features and has short-time memory function. The proposal in [26] can be good at predicting future trajectories of obstacle vehicles, but at longer trajectory data improper parameter selection can cause gradient disappearance or explosion, resulting in large errors in prediction results. The social force model is based on Newtonian dynamics and consists of expressions for individual forces that reflect the different motivations and influences of the target, which can realistically describe the movement of individuals in a group. The model in [27] can achieve high prediction accuracy by considering the target motion characteristics, but the model is computationally intensive and the accuracy of the prediction depends on ideal environments and state assumptions, which makes it difficult to achieve better prediction accuracy in practical prediction tasks.

In general, this paper proposes an innovative mathematical model of joint time-space four-dimensional imaging, based on which a BP neural network-based trajectory prediction model is proposed. Through the self-built database, both models can better achieve the prediction from the binocular pixel coordinates of the sample target points to the four-dimensional spatio-temporal coordinates. The errors of the time-space four-dimensional coordinates prediction are relatively small and therefore sufficient to predict the velocity corresponding to each moment of the target point under each kind of velocity sample, and the prediction is accurate in real time.

## 4. Conclusions

In this paper, a new time-space four-dimensional imaging model is proposed for dynamic target trajectory localization prediction. Based on the theory of ray projection and central perspective imaging model, the traditional spatial 3D imaging model was expanded along the time axis, and the time parameters and target motion parameters were written into the imaging model to establish the correspondence between the moving target object points and the sequence image points, i.e., a joint time-space four-dimensional imaging model under the moving target was established. The artificial neural network method was used to establish the prediction relationship of the time-space four-dimensional model. The four pixel coordinate points of the left and right images were used as input, and the world coordinates and time were used as output to establish the time-space four-dimensional model, and the learning and localization prediction of the motion mode were realized through the self-built database. Using the built binocular stereo vision measurement device, the data of the moving target in front were collected, and the model of this paper was used to predict the target data and realize the time-space orientation judgment of the moving target.

Based on the work in this paper, the following conclusions can be drawn.

The artificial neural network-based method can establish the mapping relationship of the time-space four-dimensional model. The experiments in this paper showed that the prediction from the binocular pixel coordinates of the sample target points to the four-dimensional time-space coordinates can be better achieved by the self-built database, regardless of the state of the moving object. The absolute errors of the spatio-temporal four-dimensional coordinates prediction were concentrated below 2.5 in the range of x and y coordinates and a thousand millimeters in the range of z coordinates, and the time range was in the unit of seconds. Moreover, there were very few errors larger than 2.5, among which the most errors were in the range of 0.5–1.5.

By use of the binocular stereo vision measurement device built in this paper, the data of the forward-moving target can be collected, and even the data that the network has never “seen” before can be input into the model of this paper to predict the target data and realize the perception and time-space orientation judgment of the moving target. From the experimental results, it can be seen that, for the depth information, when the speed was 20 mm/s the error of the network prediction value and the standard value were roughly distributed between −2.06 and 2.23, and the relative error rate was distributed between −0.18% and 0.19%; the error of the depth information prediction value for the speed group of 40 mm/s was roughly distributed between −1.94 and 2.09, and the relative error rate was distributed between −0.17% and 0.18%; The errors of the predicted depth information for the speed group of 60 mm/s were roughly distributed between −2.11 and 2.76, and the relative error rates were distributed between −0.19% and 0.23%. For the time information, the time errors were roughly distributed between −0.12 and 0.06 when the speed was 20 mm/s, and the relative error rate was distributed between −1.51% and 1.02%; when the speed was 40 mm/s, the time errors were roughly distributed between −0.09 and 0.08, and the relative error rate was distributed between −2.01% and 1.23%; when the speed was 60 mm/s, the time error was roughly distributed between −0.08 and 0.07, and the relative error rate was distributed between −1.82% and 2.03%. For the velocity information, it was shown that the network model can better predict the velocity corresponding to each moment of the target point under each velocity sample with less error.

Perhaps the recognition accuracy is not very high at this stage, but it is sufficient for the model to be effective and it can be improved by increasing the number and quality of image samples and the network structure in subsequent work. In the follow-up work, the following problems can be continued to be studied.

(1)The effectiveness of neural networks is closely related to the number and quality of training samples. Network training needs a large amount of sample data as support, but the number of extracted training samples currently relies on the traditional corner point extraction algorithm, and the accuracy of extracted target points is limited. Therefore, when studying the motion target trajectory localization prediction problem using a neural network, more accurate motion state target point data need to be collected as training samples to enhance the practicality of the algorithm.(2)The experimental device selected in this paper was a linear displacement stage, and the experimental object was to study a simple uniform linear motion target. In the subsequent research, more motion modes of the target can be explored. For example, uniform curve motion, uniform acceleration linear motion, uniform acceleration curve motion, etc. Through the enrichment of motion patterns, the algorithm can be continuously improved, while the influence on the network recognition efficiency can be further studied and discussed.(3)The output nodes of the network model in this paper were composed of x, y, z, and t. The validation calculation for velocity was obtained by predicting the results z and t. In subsequent research, whether velocity parameters, as well as multiple parameters, can be added to the neural network model, whether it is a guideline for the construction of the network structure, and how to adjust the network parameters according to the visualization results, all require further exploration.

## Figures and Tables

**Figure 1 sensors-22-06375-f001:**
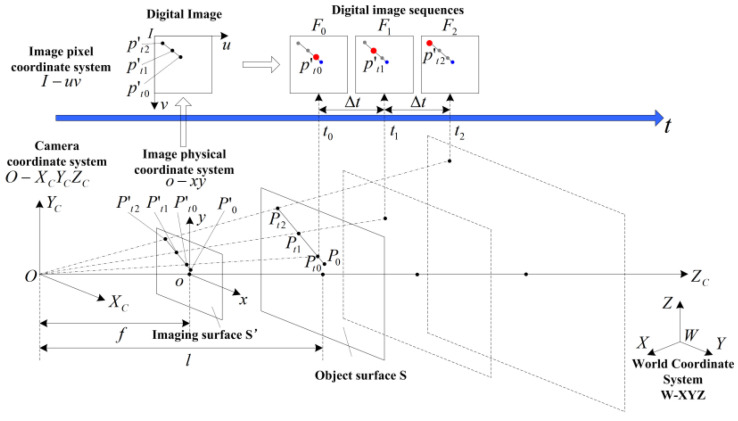
Schematic diagram of the joint time-space four-dimensional imaging model.

**Figure 2 sensors-22-06375-f002:**
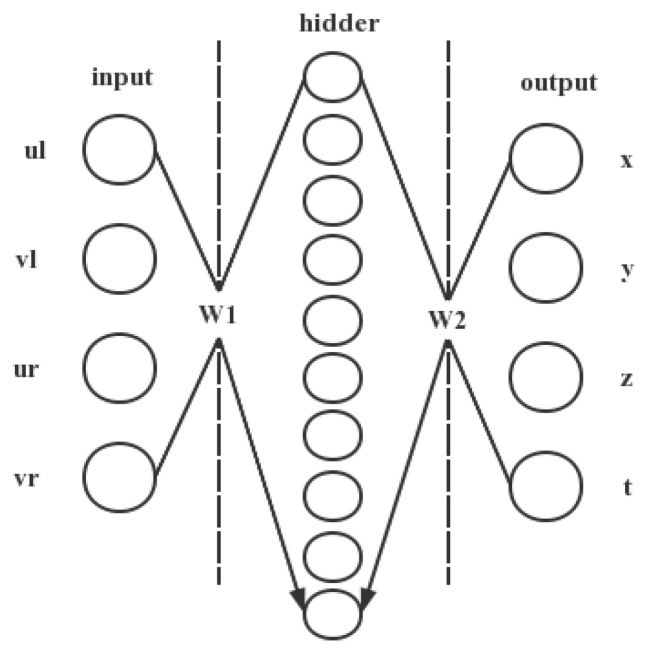
The 4-10-4 type BP network.

**Figure 3 sensors-22-06375-f003:**
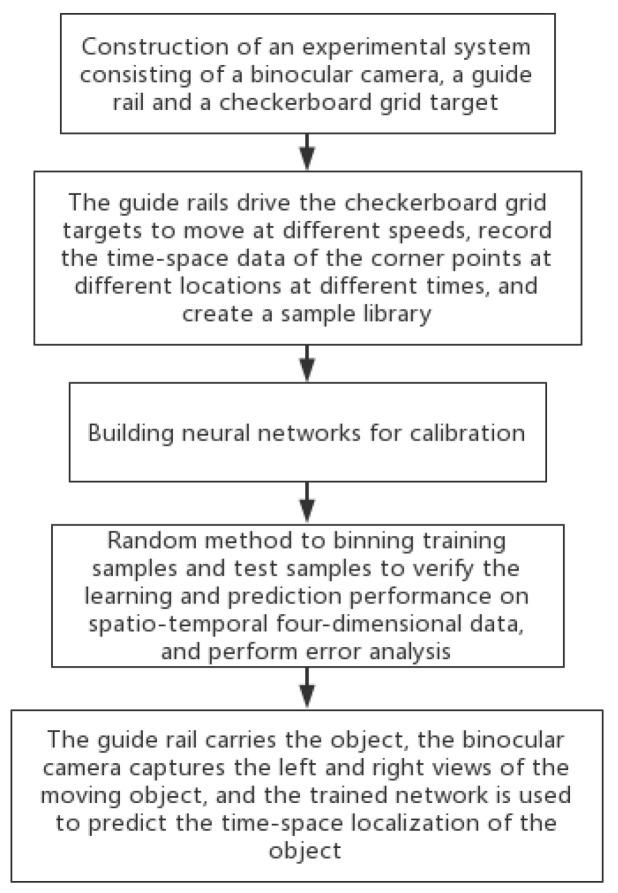
Predictive model flowchart.

**Figure 4 sensors-22-06375-f004:**
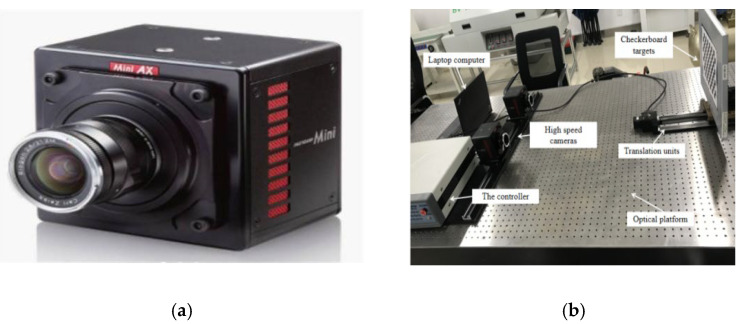
Laboratory environment map. (**a**) The high-speed camera. (**b**) Experimental system diagram.

**Figure 5 sensors-22-06375-f005:**
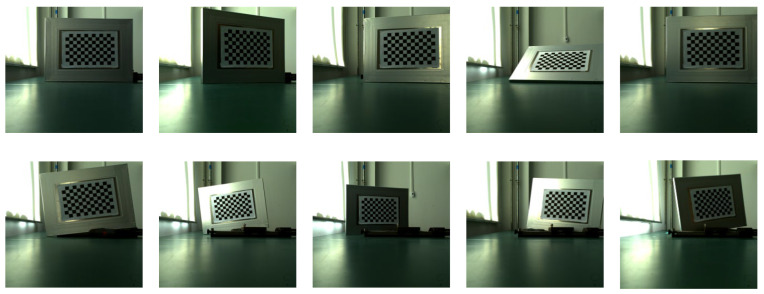
System calibration left view.

**Figure 6 sensors-22-06375-f006:**
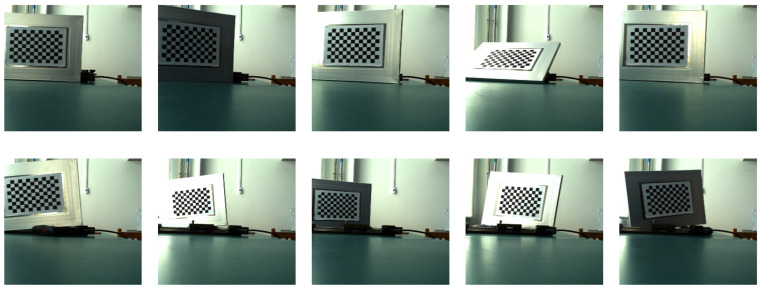
System calibration right view.

**Figure 7 sensors-22-06375-f007:**
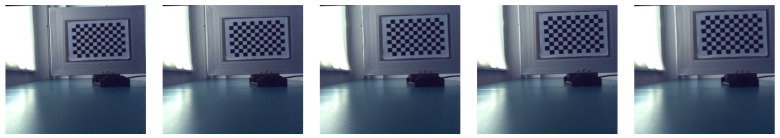
Left view of sample library production and target image acquisition.

**Figure 8 sensors-22-06375-f008:**
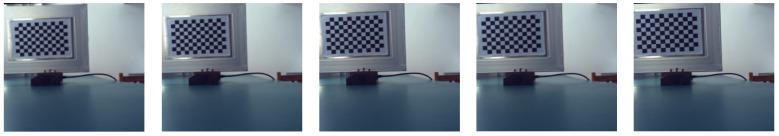
Right view of sample library production and target image acquisition.

**Figure 9 sensors-22-06375-f009:**
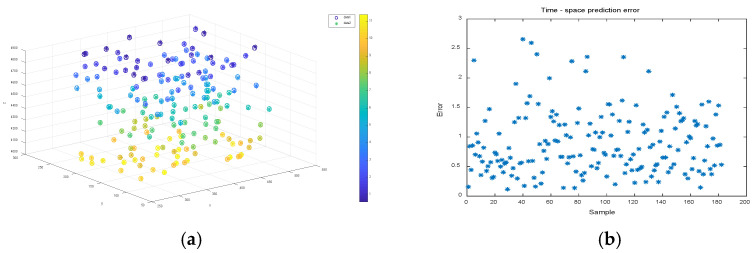
Time-space localization prediction and error of sample points for speed 20 mm/s sample bank test. (**a**) Four-dimensional positioning prediction results; (**b**) spatio-temporal positioning prediction error.

**Figure 10 sensors-22-06375-f010:**
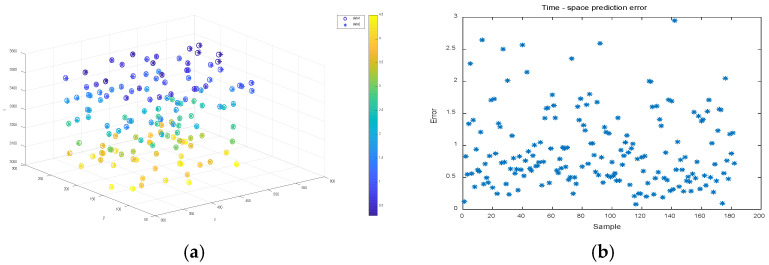
Time-space localization prediction and error of sample points for speed 40 mm/s sample bank test. (**a**) Four-dimensional positioning prediction results; (**b**) spatio-temporal positioning prediction error.

**Figure 11 sensors-22-06375-f011:**
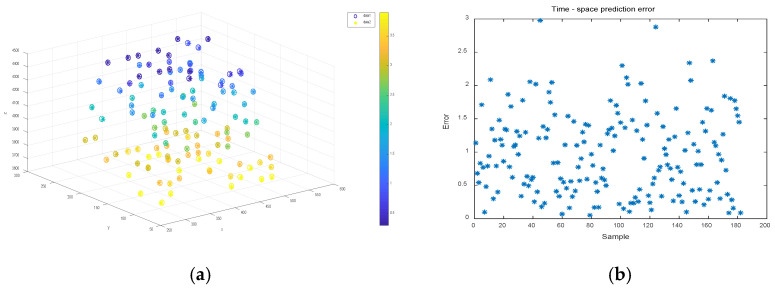
Time-space localization prediction and error of sample points for speed 60 mm/s sample bank test. (**a**) Four-dimensional positioning prediction results; (**b**) spatio-temporal positioning prediction error.

**Figure 12 sensors-22-06375-f012:**

Left view of solid motion image acquisition.

**Figure 13 sensors-22-06375-f013:**
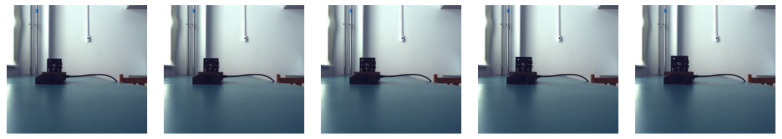
Right view of solid motion image acquisition.

**Figure 14 sensors-22-06375-f014:**
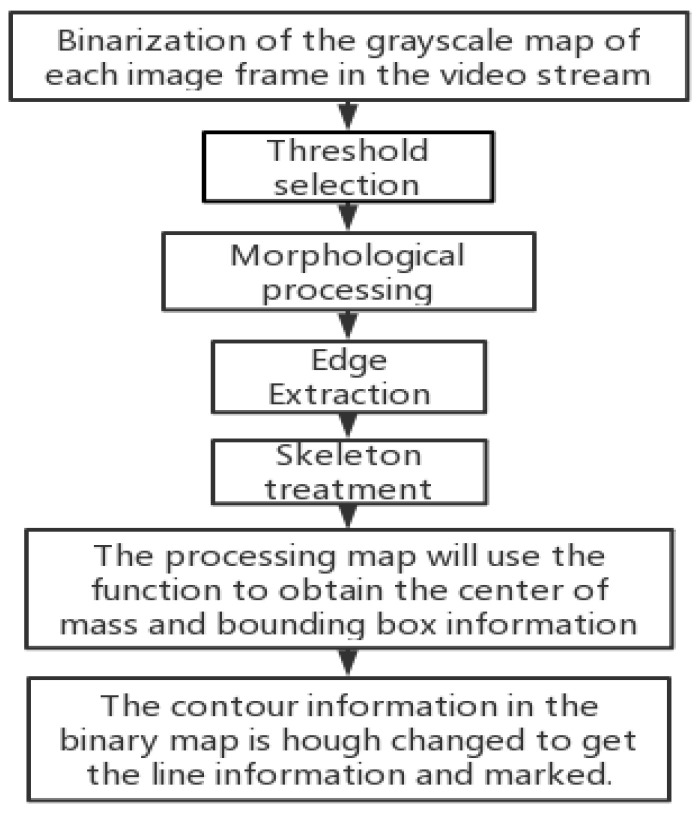
Image processing flow chart.

**Figure 15 sensors-22-06375-f015:**
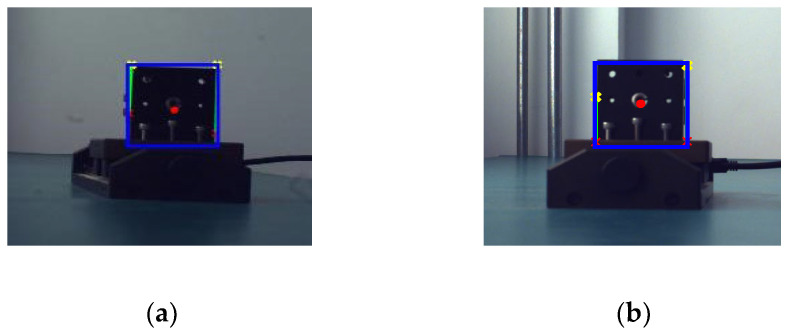
Real target bounding box (**a**) and center-of-mass point extraction (**b**).

**Figure 16 sensors-22-06375-f016:**
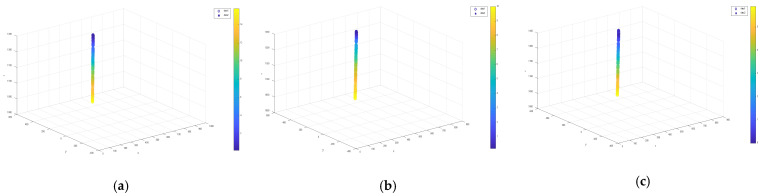
Object time−space 4D positioning prediction results. (**a**) Predicted results of object four-dimensional positioning at a speed of 20 mm/s; (**b**) predicted results of object four-dimensional positioning at a speed of 40 mm/s; (**c**) predicted results of object four-dimensional positioning at a speed of 60 mm/s.

**Figure 17 sensors-22-06375-f017:**
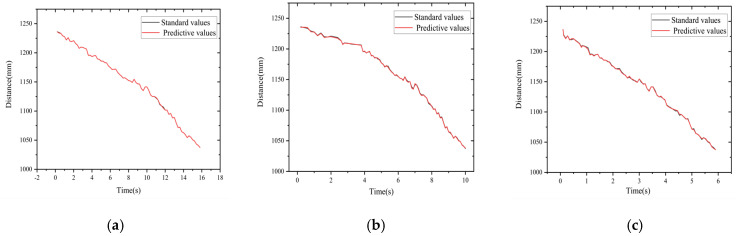
Depth− standard value vs. predicted value. (**a**) 20 mm/s. (**b**) 40 mm/s. (**c**) 60 mm/s.

**Figure 18 sensors-22-06375-f018:**
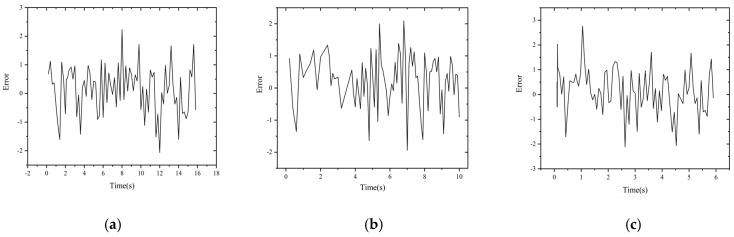
Depth− error chart. (**a**) 20 mm/s. (**b**) 40 mm/s. (**c**) 60 mm/s.

**Figure 19 sensors-22-06375-f019:**
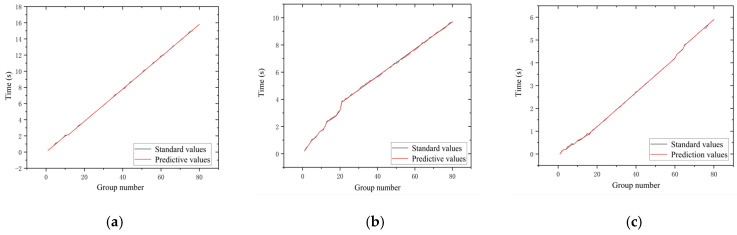
Time− standard value vs. predicted value. (**a**) 20 mm/s. (**b**) 40 mm/s. (**c**) 60 mm/s.

**Figure 20 sensors-22-06375-f020:**
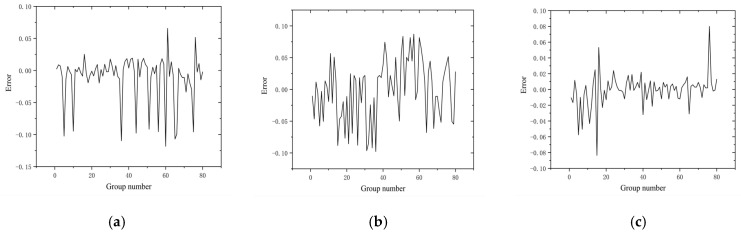
Time− error chart. (**a**) 20 mm/s. (**b**) 40 mm/s. (**c**) 60 mm/s.

**Table 1 sensors-22-06375-t001:** Neural network structure and parameter settings for the experiments in this paper.

Parameter Type	Parameter Value	Parameter Type	Parameter Value
Network structure	4–10–4	Initial weight	Nguyen–Widrow initialization
Hidden layer activation function	Tangent-sigmoid	Learning rate	0.1
Output layer activation function	Pure linear function	Training rules	Levenberg– Marquardt

**Table 2 sensors-22-06375-t002:** Hardware configuration of experimental equipment.

**High-speed camera**	**Model**		**Resolution**		**Frame Rate**	
	Photron AX200 (Manufacturer: Chengdu Guangna Technology Co., Chengdu, China)		1024 × 1024		9,000,000 fps	
**High-speed camera**	**Model**		**Resolution**		**Frame Rate**	
	Photron AX200 (Manufacturer: Chengdu Guangna Technology Co., Chengdu, China)		1024 × 1024		9,000,000 fps	
**Panning table**	**Brands**		**Model**		**Itinerary**	
	Zhuoli Hanguang		MC600-2B		200 mm	
**Controller**	**Brands**		**Model**		**Pulse Frequency**	
	Zhuoli Hanguang		MC600-2B		400k Hz	
**Co** **mputer**	**Brands**	**CPU**	**GPU0**	**GPU1**	**System**	**Learning Framework**
	Zhuoli Hanguang	Intel (R) Core (TM) i7-10750H CPU @ 2.60 GHz	Intel (R) UHD Graphics	NVIDIA Auadro T1000	Windows11	MATLAB 2017b

**Table 3 sensors-22-06375-t003:** System calibration parameters.

Calibration Parameters	Calibration Plate A	
	Left-Camera	Right-Camera
Intrinsic matrix	$\left[\begin{array}{ccc} 1753.1 & 0 & 0\\ 0 & 1749.9 & 0\\ 501.57 & 518.03 & 1 \end{array}\right]$	$\left[\begin{array}{ccc} 1758.5 & 0 & 0\\ 0 & 1759.9 & 0\\ 527.92 & 492.52 & 1 \end{array}\right]$
Rotation matrix	$\left[\begin{array}{ccc} 0.9998 & 0.0000 & 0.0192\\ -0.0002 & 1.0000 & -0.0098\\ -0.0192 & 0.0098 & 0.9998 \end{array}\right]$	
Translation matrix	$\left[\begin{array}{ccc} -271.43 & 8.6491 & 0.8022 \end{array}\right]$	

**Table 4 sensors-22-06375-t004:** Results of speed prediction error analysis.

Predicted Time $\left({\hat{t}}_{i}\right)$	Predicted Depth Value (mm) $\left({\hat{z}}_{i}\right)$	Poor Prediction Depth (mm) $\Delta {\hat{z}}_{i}=\left({\hat{z}}_{i}-{\hat{z}}_{i-1}\right)$	Predicted Time Difference (mm) $\Delta {\hat{t}}_{i}=\left({\hat{t}}_{i}-{\hat{t}}_{i-1}\right)$	Predicted Speed (mm/s) ${\hat{v}}_{i}=\frac{\Delta {\hat{z}}_{i}}{\Delta {\hat{t}}_{i}}$	Actual Speed (mm/s)	Absolute Speed Error (mm/s)	Relative Error Rate (%)
(**a**) 20 mm/s							
0.6	1243.8168	-	-	-	20	-	-
1.8	1219.5225	24.2943	1.2	20.2453		0.2453	1.23
3.0	1195.3984	24.1241	1.2	20.1034		0.1034	0.52
4.2	1171.2059	24.1925	1.2	20.1604		0.1604	0.8
5.4	1146.8428	24.3631	1.2	20.3026		0.3026	1.51
6.6	1121.7999	25.0429	1.2	20.2691		0.2691	1.35
7.8	1097.5385	24.2614	1.2	20.2178		0.2178	1.09
(**b**) 40 mm/s							
0.3	1251.8807	-	-	-	40	-	-
0.9	1227.8026	24.0781	0.6	40.1302		0.1302	0.33
1.5	1203.7593	24.0433	0.6	40.0722		0.0722	0.18
2.1	1179.7382	24.0211	0.6	40.0352		0.0352	0.08
2.7	1155.6759	24.0623	0.6	40.1038		0.1038	0.26
3.3	1131.6386	24.0373	0.6	40.0621		0.0621	0.16
3.9	1107.5944	24.0442	0.6	40.0736		0.0736	0.18
(**c**) 60 mm/s							
0.3	1281.0576	-	-	-	60	-	-
0.9	1244.9962	36.0614	0.6	60.1023		0.1023	0.17
1.5	1208.8681	36.1281	0.6	60.2135		0.2135	0.36
2.1	1172.7738	36.0943	0.6	60.1572		0.1572	0.26
2.7	1136.6973	36.0765	0.6	60.1275		0.1275	0.21
3.3	1100.5754	36.1219	0.6	60.2031		0.2031	0.39
3.9	1064.4937	36.0817	0.6	60.1361		0.1361	0.23

**Table 5 sensors-22-06375-t005:** Comparison of different models.

Algorithm	Output	Advantages	Disadvantages	Reference
Method of this paper	X, Y, Z, T	The time and space coordinates of the target point can be predicted very well, and the real-time performance is good	Rely on sample base data	-
Kalman filtering	Time	The prediction process is unbiased, stable and optimal	Poor real time, unable to predict long time	2018 [23]
BP	Longitude, Dimension	Longitudinal and latitudinal prediction of the target’s flight path can be achieved in the hotspot area	The output is the target rule trajectory, which does not reflect the real-time nature of trajectory prediction	2017 [24]
LSTM-ARIMA	Single variable	Adding feature data to the historical track data, the model effect is improved	Target movement patterns are difficult to capture in practical applications and have limitations	2019 [25]
RNN	Short- and long-term paths	Good performance in predicting the future trajectory of obstacle vehicles	Large prediction error when predicting longer trajectory data	2019 [26]
Social attention mechanism model	Accuracy	High prediction accuracy by considering target motion characteristics	Computationally intensive, relying on ideal environments and state assumptions	2020 [27]

## Data Availability

Not applicable.

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
