# Peer review of "Study on the Construction of a Time-Space Four-Dimensional Combined Imaging Model and Moving Target Location Prediction Model"

_sensors, 2022, doi:10.3390/s22176375_

Round 1

Reviewer 1 Report

1. Figure 1 needs high resolution. Please redraw this figure.

2. The proposed BP ANN is so small, I think you should try more complex NN models, at least CNN networks.

[1] Q. Huang, et al., "Development of CNN-based visual recognition air conditioner for smart buildings", Journal of Information Technology in Construction, vol. 25, pp. 361-373, 2020.

3. You may also think about memory-reduction methods for neural networks, such as network pruning, weight quantization, transfer learning, etc.

[1] Q. Huang, "Weight-Quantized SqueezeNet for Resource-Constrained Robot Vacuums for Indoor Obstacle Classification", AI, 3(1), pp. 180-193, 2022.

[2] Z. Tang et al., "Automatic Sparse Connectivity Learning for Neural Networks," in IEEE Transactions on Neural Networks and Learning Systems, doi: 10.1109/TNNLS.2022.3141665.

[3] J. Zheng, C. Lu, C. Hao, D. Chen and D. Guo, "Improving the Generalization Ability of Deep Neural Networks for Cross-Domain Visual Recognition," in IEEE Transactions on Cognitive and Developmental Systems, vol. 13, no. 3, pp. 607-620, Sept. 2021, doi: 10.1109/TCDS.2020.2965166.

4. Please add a table in the experimental result section to compare this work with existing works in the literature.

Reviewer 2 Report

This paper describes a geometric interpretation for 4-dimensional target location prediction. 

In Section 2.1, 4-dimensional time-space model is well described. The coordinate system transformation is well explained. The complex system is described mathematically.

In Section 2.2, backpropagation algorithm for general neural networks is described. However, the authors just described the classic backpropagation algorithm only. The forward propagation and backward propagation with differentiation equations are show, but those equations are almost exactly appeared in many conventional papers and textbooks already, and there is no novel aspects for the given problem. Moreover, the specific network architecture for the given target location is not shown. Figure 2 seems to be general network, but not for the proposed method. 

In Section 2.3, it is hard to understand the objective of the forecasting process. Is it just an affine transform?

The reviewer suggests the authors Section 2.2 and 2.3 to explain the specific network architecture for the proposed method. 

Round 2

Reviewer 1 Report

The quality of this work has been improved 

Reviewer 2 Report

The authors have made appropriate replies and improvements in the revised version. The proposed method does not focus on the novel neural network models, so Section 2.2 does not explain new models but the conventional NN training algorithm. Few more suggestions for the revised version:

1. The name of Section 2.2 is somewhat misleading. Suggest simply "Prediction model". 

2. Equation 13 is wrong. One of the possible corrections:

E^P -> E^2

If this correction is right, E_RMSE is better than E_RME (never heard of it).

3. Page 7, line 261

- What is E_MIN ?  Not explained.

- (typo?) "<E_RME < E_MIN" --> "E_RME < E_MIN" ?

4. The training algorithm in Equations 14-17 is a special case for sigmoid activation function in Equation 11.  Please specify that sigmoid activation function is used, near Equation 11 or 14.
